# Mutation Profile of Aggressive Pheochromocytoma and Paraganglioma with Comparison of TCGA Data

**DOI:** 10.3390/cancers13102389

**Published:** 2021-05-14

**Authors:** Yun Mi Choi, Jinyeong Lim, Min Ji Jeon, Yu-Mi Lee, Tae-Yon Sung, Eun-Gyoung Hong, Ji-Young Lee, Se Jin Jang, Won Gu Kim, Dong Eun Song, Sung-Min Chun

**Affiliations:** 1Department of Internal Medicine, Hallym University Dongtan Sacred Heart Hospital, Hallym University College of Medicine, Gyeonggi-Do 18450, Korea; ymchoi@hallym.or.kr (Y.M.C.); hegletter@hallym.or.kr (E.-G.H.); 2Asan Center for Cancer Genome Discovery, Asan Institute for Life Sciences, Seoul 05505, Korea; aster1217@gmail.com (J.L.); jangsejin@amc.seoul.kr (S.J.J.); 3Department of Internal Medicine, Asan Medical Center, University of Ulsan College of Medicine, Seoul 05505, Korea; mj080332@amc.seoul.kr (M.J.J.); wongukim@amc.seoul.kr (W.G.K.); 4Department of Surgery, Asan Medical Center, University of Ulsan College of Medicine, Seoul 05505, Korea; niphredil@amc.seoul.kr (Y.-M.L.); tysung@amc.seoul.kr (T.-Y.S.); 5Department of Medical Science, Asan Medical Center, Asan Medical Institute of Convergence Science and Technology, University of Ulsan College of Medicine, Seoul 05505, Korea; easy0@hanmail.net; 6Department of Pathology, Asan Medical Center, University of Ulsan College of Medicine, Seoul 05505, Korea

**Keywords:** aggressive pheochromocytoma, paraganglioma, DNA mutation analysis, high-throughput nucleotide sequencing, prognosis

## Abstract

**Simple Summary:**

Pheochromocytomas and paragangliomas (PPGLs) are neuroendocrine tumors arising from chromaffin cells of the adrenal medulla, or extra-adrenal paraganglia, respectively. In PPGLs, germline or somatic mutations in one of the known susceptibility genes are identified in up to 60% patients. Recent WHO classification defines that all PPGLs can have metastatic potential. The term, ‘malignant’ is replaced with ‘metastatic’ in this group of tumors. However, the peculiar genetic events that drive the aggressive behavior, including metastasis in PPGLs are yet poorly understood. We performed targeted next-generation sequencing analysis to characterize the mutation profile in fifteen aggressive PPGL patients and compared accessible data of aggressive PPGLs from The Cancer Genome Atlas (TCGA) with findings of our cohort. This targeted mutational analysis might expand the mutation profile of aggressive PPGLs, and may also be useful in detecting the possible experimental therapeutic options or predicting poor prognosis.

**Abstract:**

In pheochromocytoma and paraganglioma (PPGL), germline or somatic mutations in one of the known susceptibility genes are identified in up to 60% patients. However, the peculiar genetic events that drive the aggressive behavior including metastasis in PPGL are poorly understood. We performed targeted next-generation sequencing analysis to characterize the mutation profile in fifteen aggressive PPGL patients and compared accessible data of aggressive PPGLs from The Cancer Genome Atlas (TCGA) with findings of our cohort. A total of 115 germline and 34 somatic variants were identified with a median 0.58 per megabase tumor mutation burden in our cohort. The most frequent mutation was *SDHB* germline mutation (27%) and the second frequent mutations were somatic mutations for *SETD2*, *NF1*, and *HRAS* (13%, respectively). Patients were subtyped into three categories based on the kind of mutated genes: pseudohypoxia (*n* = 5), kinase (*n* = 5), and unknown (*n* = 5) group. In copy number variation analysis, deletion of chromosome arm 1p harboring *SDHB* gene was the most frequently observed. In our cohort, *SDHB* mutation and pseudohypoxia subtype were significantly associated with poor overall survival. In conclusion, subtyping of mutation profile can be helpful in aggressive PPGL patients with heterogeneous prognosis to make relevant follow-up plan and achieve proper treatment.

## 1. Introduction

Pheochromocytomas and paragangliomas (PPGLs) are neuroendocrine tumors arising from chromaffin cells of the adrenal medulla or extra-adrenal paraganglia, respectively [1,2]. PPGLs are now recognized to have the highest degree, up to approximately 40%, of heritability of any tumor type [3]. Mutations in more than 19 genes were discovered to be involved in tumorigenesis, of which at least 12 are associated with a syndromic PPGL presentation [4,5,6]. More recently, somatic mutations began to be uncovered in PPGLs without a recognizable germline variant and estimated to be detected in an additional 25–30% of tumors [7,8,9,10,11,12,13,14,15,16]. In addition, to derive a molecular classification for PPGLs, The Cancer Genome Atlas (TCGA) group performed unsupervised clustering of tumor mRNA expression profiles, detecting four significant expression subtypes: Kinase signaling, pseudohypoxia, *Wnt*-altered, and cortical admixture [15].

The diagnosis of metastatic PPGL is made upon detection of metastatic tumor spread in sites where chromaffin cells are normally absent [1,17,18]. A recent WHO classification explains that all pheochromocytomas can have metastatic potential. The term, ‘malignant’, is replaced with ‘metastatic’ in this group of tumors [1]. However, the genetic events that drive the metastatic events of PPGL are yet poorly understood [5,19]. *SDHB* gene was reported to confer a higher risk of malignancy [20,21], but only half of patients with metastatic disease carried inherited *SDHB* mutations [22]. Recently, germline *FH* mutation, *MAX* mutation, *MAML3* fusion, *ATRX*, and *SETD2* genes were identified as a novel factor associated with tumor aggressiveness [15,19,23,24,25]. These new findings may lead to understanding of its underlying biology and may give insight of potential new pathways for personalizing molecular targeted therapies in metastatic PPGL [7].

In this study, we analyzed the mutation profile of aggressive PPGLs using targeted next-generation sequencing (NGS) analysis and compared with clinically aggressive PPGLs of TCGA data to characterize the genetic events associated with the aggressive behavior, including metastasis in PPGLs.

## 2. Materials and Methods

### 2.1. Study Subjects and Tissue Sample

Among the previously reported PPGL series from Asan Medical Center (AMC), Seoul, Korea [26], fifteen aggressive patients (11 pheochromocytomas and 4 sympathetic paragangliomas) with available formalin-fixed, paraffin embedded (FFPE) archival samples were enrolled for targeted NGS in this study. Fifteen samples from aggressive (4 regional and 11 distant metastasis) patients corresponded to 8 primary tumors and 7 metastases. Targeted NGS analysis with custom designed panel was performed with tumor and matched normal mode for all patient samples except one (a normal sample of patient_13) due to fail of quality check. All pathological specimens were reviewed by an endocrine pathologist (D.E.S.). The pathologist selected adequate tissue blocks and marked tumor area to determine tumor purity and to guide isolation of DNA from PPGL tissues and matched normal tissues. The protocol of this study was approved by the Institutional Review Board of the AMC. Written informed consents were obtained from all patients who were alive.

### 2.2. DNA Extraction

Depending on the sample size and tumor cellularity, genomic DNA was extracted from two to five 6-μm-thick slices per tumor or matched normal FFPE tissue sample. After de-paraffinization with xylene and ethanol, genomic DNA was isolated with a NEXprep FFPE Tissue Kit (#NexK-9000; Geneslabs, Seongnam, Korea) according to the manufacturer’s recommendations. Briefly, tissue pellets were completely lysed with proteinase K in lysis buffer overnight at 56 °C, and followed by an additional incubation for 3 min with magnetic beads and Solution A at room temperature. After incubation for 5 min on a magnetic stand, the supernatants were removed, and remaining beads were washed three times with ethanol to remove residual contaminants and dried for 5 min. Finally, genomic DNA was eluted in 50 μL of DNase free water and quantified using the Qubit™ dsDNA HS Assay kit (Thermo Fisher Scientific, Waltham, MA, USA).

### 2.3. Targeted Next-Generation Sequencing (NGS) and Data Processing

A DNA library was prepared as described in our previous report using the S1 method [27]. Briefly, gDNA shearing with S1 enzyme, end repair, A-tailing, and ligation with the TruSeq adaptor using the SureSelect XT reagent kit (Agilent Technologies, Santa Clara, CA, USA) was performed. Each library constructed with sample-specific unique sequencing barcode (6 bp) was quantified using the Qubit kit, then four libraries were pooled (yielding a total of 720 ng) for target capture using the Agilent SureSelectXT custom kit (OP_v2 RNA bait, 2.9 Mb; Agilent Technologies). OP_v2 panel was designed to capture the exons of 505 cancer-related genes plus partial introns from 15 genes often rearranged in cancer (Appendix A). The captured libraries were enriched by limited PCR (10 cycles) and measured by the Qubit kit. DNA libraries that passed quality checks were sequenced using MiSeq platform (Illumina, San Diego, CA, USA) for paired-end sequencing. Sequenced reads were aligned to the human reference genome (NCBI build 37) with Burrows-Wheeler Aligner (0.5.9) with default options, and PCR de-duplication was performed using a Picard’s MarkDuplicates package. After initial alignment process, reads were realigned at common indel positions with GATK IndelRealigner, and then recalibration of base quality was done using the GATK Table-Recalibration, and used as final BAM.

### 2.4. Variant Calling and Filtering

For accurate detection of somatic and germline variants, final BAM files with the recalibrated base quality were used. Germline variants were initially called using the GATK’s Haplotypecaller, CNNscoreVariants in single sample mode and additional filtering was performed with FilterVariantTranches in default options to discard frequent false positive variation. Somatic variants of tumor DNA were called with matched normal DNA using the MuTect2 (4.1.7). After additional filtering using GATK4 FilterMutectCalls tools, final somatic variants were annotated using the Ensemble Variant Effect Predictor (version 86) and were then converted to the Mutation Annotation Format (MAF) file using vcf2maf (https://github.com/mskcc/vcf2maf, v1.6.17, accessed on 23 July 2020). Further, manual curation using the Integrative Genomics Viewer (IGV) was performed for filtering false-positive variants. The list of variants was additionally annotated clinically actionable mutation and clinical significance by OncoKB [28] and Clinvar database [29]. For patient #13, without a matched normal, variant allele fraction of tumor DNA was used to infer germline variants. Three variants representing 50% of variant allele fraction were presumed as germline mutations.

### 2.5. Copy Number Variants Analysis

Somatic copy number variations were evaluated using CNVKit (0.9.6). Final BAM files of each tumor DNA were used as input files for CNVkit using default parameters with reference file generated by the pooled normal samples. The log2 values of each bins from CNR file were used for downstream CNV analysis. Each bins with log2 value higher than 0.8 or less than −0.8 were classified as amplification, and deletion, respectively. CNV analysis was performed on only genes with sufficient coverage depth to remove false positive one due to capture efficiency issue.

### 2.6. Tumor Mutation Burden Analysis

For the analysis of tumor mutation burden, we used TCGA cancer genomics dataset across 33 different cancer types from https://gdc.cancer.gov/about-data/publications/mc3-2017 (mc3-2017, accessed on 28 July 2020). Among the TCGA’s PPGL cohort, we classified 16 clinically aggressive samples as aggressive subjects based on clinical information from previous reports, including having distant metastatic events, positive local lymph nodes or local recurrence [15], and the rest were assumed as non-aggressive subjects.

### 2.7. Statistical Analysis

R version 4.0 and R libraries prodlim, car, Cairo, and survival were used for analyzing data and drawing graphs (R foundation for Statistical Computing, Vienna, Austria, http://www.R-project.org, accessed on 13 May 2020). Survival curves were constructed using the Kaplan–Meier method, and the log-rank test was used to evaluate differences in survival according to genetic alterations. A Cox proportional hazards model with hazard ratios (HRs) and 95% confidence intervals (CIs) was used to evaluate the risk of death or recurrence. All *p*-values were two sided, with *p* < 0.05 considered statistically significant.

## 3. Results

### 3.1. Baseline Characteristics

Table 1 presents the baseline characteristics of the 15 AMC cohort subjects and 16 aggressive TCGA subjects. The median age of the patients was 37.8 and 46.0 years, and 80% and 50% of the patients were females in AMC, and TCGA data, respectively. Distant metastasis either synchronous or metachronous has occurred in 11 (73.3%) and 11 (68.8%) subjects, respectively. Other 4 patients in AMC had local lymph nodes, and other 5 patients in TCGA cohort were subjects with positive lymph nodes or local recurrence. The AJCC staging was applied as revised staging after recurrence or metastasis. Clinical and pathological characteristics of 15 aggressive AMC subjects were presented in Appendix A.

### 3.2. Characteristics of Mutation Profile of AMC Cohort

The overall mean target coverage and percentage of bases covered >30× for the 15 tumor DNA were 63× depths and 80%, respectively. Total 115 germline and 34 non-synonymous somatic variants were identified. For somatic variants, 26 coding missense mutations, 5 frameshift deletions, 2 frameshift insertions, and one mutation at splicing region were identified.

Genes with two or more mutation frequency were depicted in Figure 1A and Table 2. The most frequent variation was germline mutation of *SDHB* gene (mutation frequency 4, 27%). Somatic mutations were frequently identified in the *SETD2, NF1*, and *HRAS* genes (mutation frequency 2, 13%, respectively) (Figure 1B). One sample with somatic *SDHB* mutation was accompanied by germline *TP53* mutation. One familial multiple endocrine neoplasia type 2 (MEN2) sample (Patient_15) revealed an expected germline *RET* mutation. One sample (Patient_1) showed multi-hit *SEDT2* mutations, one germline and the other somatic mutation. One sample (Patient_2) had germline *SDHB* mutation together with somatic *NF1* mutation.

### 3.3. Classification of PPGL Based on the Pattern of Mutation Profile

Aggressive TCGA subjects were sorted to pseudohypoxia (*n* = 8, 50%), *Wnt*-altered pathway (*n* = 6, 38%), and kinase signaling pathway (*n* = 2, 13%) based on previous mRNA subtypes classifications [15] (Figure 2B).

As altered genes, including both germline and somatic mutation were subtyped based on these four pathways, AMC aggressive subjects could be sorted to pseudohypoxia group (*n* = 5) and kinase group (*n* = 5) (Figure 2A). Since the targeted NGS panel, used in this study, did not cover *CSDE1*, *EPAS1*, *MAX* mutations and *MAML* fusions which can be markers of *Wnt*-signaling, pseudohypoxia or cortical admixture pathways, some samples could not be further distinguished. The samples which cannot be further classified as unknown subtype (*n* = 5) were defined. Among them, two samples showed *ATRX* mutations (one germline and the other somatic). One sample (Patient_4) with somatic *ATRX* mutation was accompanied by somatic *SETD2* mutation. Another one sample (Patient_5) with germline *ATRX* mutation and germline *KMT2C* mutation was detected. Additionally, we performed structural variation and fusion analysis, but there was no significant alteration.

### 3.4. Copy Number Variation

NGS-based copy number variant (CNV) detection for each analyzed tumor sample is presented in Figure 3A. The most frequently observed pattern was deletion of chromosome arm 1p where *SDHB* is present. Those who had no deletion in chromosome arm 1p (patient_1, 6, 14) showed complete response according to RECIST criteria. When the copy number variation analysis was performed by specific gene segmentation, *SDHB* deletion was the most frequently detected variation with additional deletions in *ATRX*, *SDHD*, *NF1*, and *VHL* genes. Amplification of *SDHC* gene was also detected (Figure 3B).

### 3.5. Tumor Mutation Burden

Tumors of AMC aggressive cohort had median 0.58 (0.58–4.11) per megabase tumor mutation burden (TMB), which was defined as counting the number of nonsynonymous mutations found only within the tumor sample. TMBs of TCGA non-aggressive cohort (*n* = 156) and aggressive cohort (*n* = 16) were 0.18, and 0.32, respectively. The captured coding region covered 1.7 Mb in AMC cohort and 50 Mb in TCGA cohort. Figure 4 shows TMB of our aggressive PPGL subjects in comparison with various other cancer types.

### 3.6. Survival Analysis

The overall median survival of AMC aggressive subjects was 15.64 years, and that of TCGA subjects was not defined (Figure 5A). The median disease free survival of 12 AMC subjects with metachronous metastatic presentation was 5.13 years and that of aggressive TCGA subjects was 1.95 years (Figure 5B). There was no difference in both overall and disease free survival between AMC and aggressive TCGA cohort.

We did survival analysis of AMC aggressive cohort according to the presence of germline or somatic *SDHB* mutation (Figure 5C). The risk of death was significantly increased in patients with *SDHB* mutation (HR = 4.66, 95% CI 1.10–19.75, *p* = 0.04) compared to patients without this mutation. On the other hand, disease free survival was not significantly different according to the presence of *SDHB* mutation (HR = 2.10, 95% CI 0.55–7.94, *p* = 0.28) (Figure 5D).

In addition, we did survival analysis of AMC aggressive cohort according to the mutation profile subtypes. The pseudohypoxia subtype was significantly associated with poor prognosis. Kaplan–Meier plots revealed that there were fewer deaths (*n* = 0) in patients with kinase subtype compared with patients with pseudohypoxia or unknown subtypes (*p* = 0.02 by log-rank test, Figure 5E). The disease free survival was not significantly different according to the mutation profile subtypes (Figure 5F).

## 4. Discussion

In this study, based on the targeted NGS analysis, 15 AMC aggressive PPGL subjects showed a lower somatic mutation frequency in comparison with other cancers. The median TMB was found to be 0.58/Mb, which is slightly higher than TMB of TCGA aggressive dataset. This difference appears to be due to use of a targeted NGS panel focused on cancer-related genes compared to the whole exome sequencing (WES) panel used in the TCGA dataset. In AMC cohort, there was one familial MEN2 case with germline *RET* mutation. Metastatic pheochromocytoma related with MEN2 is rare and the rate is estimated to be only 3–4% of cases [30,31]. In all 14 apparently sporadic PPGLs, several germline and somatic mutations had been uncovered in various genes including previously reported as PPGL related genes [4,5,12,13,15]. Based on the mutation patterns, each five patients were grouped into pseudohypoxic, kinase, and unknown subtypes in our aggressive PPGLs. In unknown subtype, several alterations of chromatin remodeling genes, including *ATRX*, *SETD2*, *KMT2C* and *SMARCA4* were found up to 60%. We confirmed again pseudohypoxia subtype showed significantly poor prognosis compared to other subtypes.

Recently, three sets of gene panels were proposed by NGSnPPGL (NGS in PPGL study group) [32]. The known PPGL-related genes according to this consensus statement, but not included in this panel are listed in Appendix A.

Given the heterogeneous prognosis of aggressive PPGLs, many previous studies suggested comprehensive analysis including variable clinical prognostic markers, like tumor location, size, and hormone secretion [26,33]. Recently, some papers regarding genetic markers for prognostic indicators were published, but no clear results for the prediction of metastasis were established so far.

In this study, four germline and rare one somatic *SDHB* mutations were detected in five patients. Cancer predisposition associated with *SDHB* germline mutation in PPGL patients is well known, however somatic *SDHB* mutation had been rarely reported [34]. Recently, *SDHB* and *SDHD* somatic mutations were reported, and all in metastatic PPGL cases [35,36]. This somatic *SDHB* mutation was accompanied by germline *TP53*, the tumor suppressor gene mutation in our cohort. Another aggressive metastatic PPGL case with concurrent germline *SDHB* and *TP53* mutation was previously reported [37]. These *SDHB*-mutated pseudohypoxic subtype PPGL subjects showed much shorter overall survival in this study, as it is known [38]. Meanwhile, another previous study did not show any prognostic role of *SDHB* mutation [33]. This discrepancy between various studies might be caused by many confounding other clinical factors, which affect the aggressive behavior of PPGLs.

Previously, kinase subtype has not been considered to represent aggressive PPGLs [3]. However, in this study, one third of aggressive PPGLs in AMC samples were classified as kinase subtype whereas only 12% of aggressive TCGA cohort was kinase group. Among five kinase group subjects in AMC, three had distant metastasis and two revealed regional metastasis. Those kinase subtype PPGLs revealed much better survival than pseudohypoxic subtype in AMC aggressive cohort. Further extensive genetic studies are needed to elucidate the prognostic implication of metastatic PPGLs in this kinase subtype.

In this study, five samples were defined as mutation profile of unknown subtype. Among them, three samples (60%) showed various alteration in chromatin-remodeling genes including *ATRX*, *SETD2*, *KMT2C* and *SMARCA4*. These chromatin-remodeling genes were previously identified in PPGLs [14,15,39,40,41]. Somatic *ATRX* mutations in PPGLs were recently reported to be associated with malignant behavior of these tumors [23,34,39,41]. It has been repeatedly identified together with *SDHB* mutations [35,39], but both *ATRX* mutations in our cohort were not accompanied by *SDHB* mutation. *SETD2* gene was also previously mentioned as being associated with aggressive behavior in patients with PPGLs [19]. A further genetic study is required to elucidate the late role of these altered chromatin-remodeling genes in the disease progression of aggressive PPGLs patients.

This study has several limitations. First, there is no true validation cohort to prove the presence of similar mutations in another aggressive PPGLs cohort or to investigate general prevalence of the altered mutations. Furthermore, we did not include control group of non-metastatic PPGLs. Second, this cohort is not entirely composed of metastatic PPGLs, and it is hard to find out the characteristic mutations predicting metastatic potential. Third, we evaluated limited number of already known PPGL related genes using a targeted NGS panel. Fourth, a combined analysis of biochemical phenotype was not performed to do genotype-phenotype correlation. Fifth, our cohort sample size is too small to derive proper statistical significance.

Although there was rapid progress for clarification of various germline or somatic genetic alterations in PPGLs, very little is known about the peculiar genetic changes associated with aggressive metastatic behavior. Importantly, early prediction of metastatic potential can enable the effective decision making for the relevant follow-up plan. Furthermore, aggressive PPGLs represent a major clinical challenge in choosing a proper clinical treatment. We proved the poor overall survival in aggressive PPGLs patients with germline and rare somatic *SDHB* mutation and mutational profile of pseudohypoxia subtype using targeted NGS analysis. We suggested possible synergistic effect of various chromatin remodeling genes alterations for the progression of PPGLs in the unknown subtype. Further studies using more comprehensive genomic profiling are urgently needed to expand the possible number of experimental therapeutic options for aggressive PPGLs patients.

## 5. Conclusions

In this study, we confirmed again worse prognostic impact of pseudohypoxia subtype or *SDHB* mutations compared to relatively better prognosis of kinase subtype. Moreover, we supported the potential role of additional genetic alterations of chromatin remodeling genes including *SETD2*, *ATRX*, *KMT2C* and *SMARCA4* through the unknown mutation profile group for the progression of PPGLs. We recommend subtyping of the mutation profile to be performed, including both germline and somatic alterations for all PPGL patients based on the heterogeneous prognosis to make relevant follow-up plan and achieve proper treatment.

## Figures and Tables

**Figure 1 cancers-13-02389-f001:**
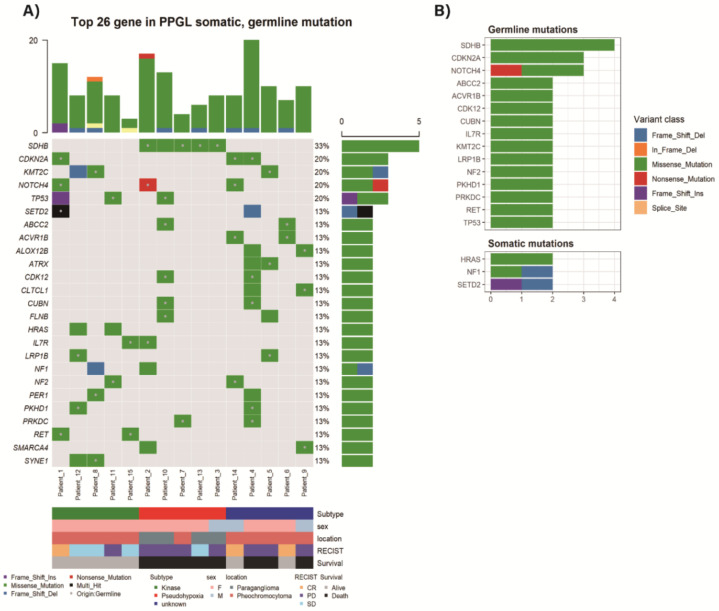
Top 26 altered genes in Asan Medical Center (AMC) pheochromocytomas and paragangliomas (PPGLs) with germline or somatic mutation (**A**) Mutations detected by targeted massive parallel sequencing were depicted. Tumor samples are arranged from left to right. The type of mutation is annotated for each sample by the color. Germline mutation is marked as a dot in a square. The mutation frequency is presented in the right of the panel. The mutation number per sample is presented on the top of the panel. (**B**) Most frequently altered germline and somatic mutations.

**Figure 2 cancers-13-02389-f002:**
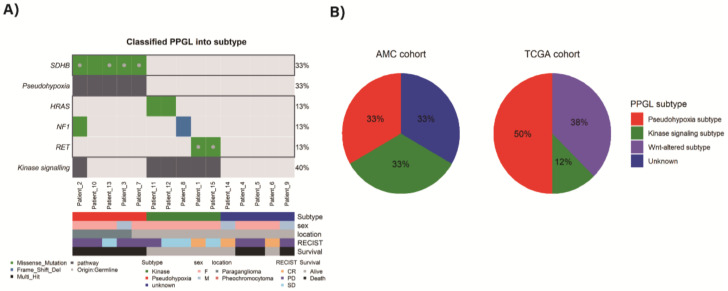
(**A**) Classified mutation subtypes of AMC PPGLs, (**B**) proportion of mutation subtypes in both AMC and TCGA cohorts.

**Figure 3 cancers-13-02389-f003:**
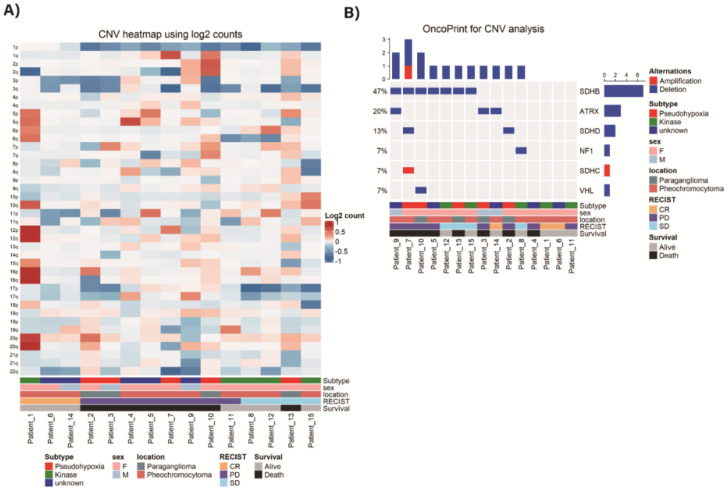
Next-Generation Sequencing (NGS)-based copy number variant (CNV) detection. (**A**) CNV heatmap using log2 counts. (**B**) CNV analysis of specific genes.

**Figure 4 cancers-13-02389-f004:**
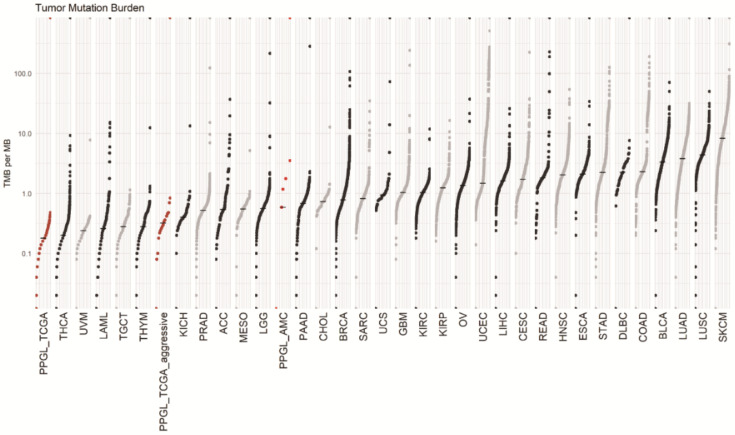
The tumor mutation burden (TMB) of metastatic pheochromocytoma and paraganglioma (PPGL) of AMC was 0.58. That of TCGA non-aggressive and aggressive tumor was 0.18 and 0.32 respectively. Data source: https://gdc.cancer.gov/about-data/publications/mc3-2017, accessed on 28 July 2020. Abbreviations for tumors and median TMB are summarized in Appendix A.

**Figure 5 cancers-13-02389-f005:**
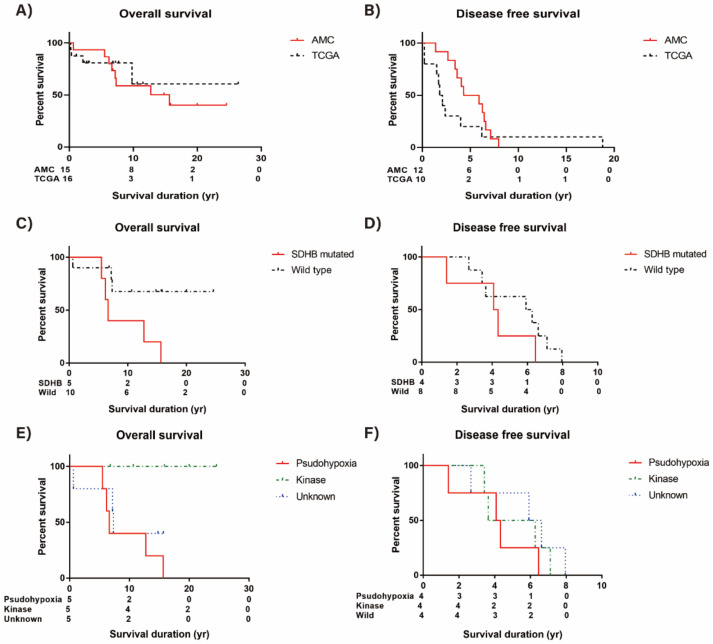
Survival of aggressive pheochromocytoma and paraganglioma (PPGL). (**A**) Overall survival (OS) of AMC and TCGA cohort, (**B**) Disease free survival (DFS) of AMC and TCGA cohort, (**C**) OS difference of AMC subjects according to *SDHB* mutation, (**D**) DFS difference of AMC subjects according to *SDHB* mutation, (**E**) OS difference of AMC subjects according to mutation subtypes, and (**F**) DFS difference of AMC subjects according to mutation subtypes.

**Table 1 cancers-13-02389-t001:** Baseline Characteristics.

		AMC Data (*N* = 15)	TCGA Data (*N* = 16)
Age, median [IQR] (year)		37.8 [30.9–48.3]	46.0 [42–62.3]
Sex (Female), *N* (%)		12 (80.0)	8 (50.0)
Hereditary case, *N* (%)		1 (6.7)	0 (0)
Tumor location (Adrenal PCC)		11 (73.3)	10 (62.5)
Functioning tumor, *N* (%)		13 (86.7)	12 (92.3)
AJCC staging ^a^	I–II	0	0
	III	4 (26.7)	5 (31.3)
	IV	11 (73.3)	11 (68.8)
Distant metastasis		11 (73.3)	11 (68.8)
RECIST	CR	3 (20.0)	NA
	PR	0	NA
	SD	4 (26.7)	NA
	PD	8 (53.3)	NA
Survival status (Death)		8 (53.3)	4 (25.0)

N, number; AMC, Asan Medical Center; TCGA, The Cancer Genome Atlas; PCC, pheochromocytoma; AJCC, American Joint Committee on Cancer; RECIST, Response evaluation criteria in solid tumors; CR, complete response; PR, partial response; SD stable disease; PD, progressive disease. ^a^ Revised final staging.

**Table 2 cancers-13-02389-t002:** Mutations identified in 15 AMC aggressive pheochromocytomas and paragangliomas.

	#Pt	Gene Symbol	HGVS.codon	HGVS.protein	Mutation Class	Clinically Actionable Mutation	Clinical Significance
1	#6	ABCC2	c.3436C > T	p.Arg1146Cys	Germline	NA	Uncertain significance
2	#10	ABCC2	c.1177C > T	p.Arg393Trp	Germline	NA	Likely pathogenic
3	#14	ACVR1B	c.865C > T	p.Pro289Ser	Germline	NA	NA
4	#6	ACVR1B	c.865C > T	p.Pro289Ser	Germline	NA	NA
5	#4	ALOX12B	c.1496G > A	p.Arg499His	Somatic	NA	NA
6	#9	ALOX12B	c.1643G > A	p.Arg548Gln	Germline	NA	NA
7	#4	ATRX	c.5400G > T	p.Met1800Ile	Somatic	NA	NA
8	#5	ATRX	c.2965A > G	p.Thr989Ala	Germline	NA	NA
9	#4	CDK12	c.404A > G	p.Glu135Gly	Germline	NA	NA
10	#10	CDK12	c.2089C > T	p.Pro697Ser	Germline	NA	NA
11	#14	CDKN2A	c.315C > A	p.Asp105Glu	Germline	NA	Uncertain significance
12	#4	CDKN2A	c.197A > G	p.His66Arg	Germline	NA	Likely benign
13	#1	CDKN2A	c.496C > T	p.His166Tyr	Germline	NA	Not provided
14	#4	CLTCL1	c.4597C > T	p.Leu1533Phe	Somatic	NA	NA
15	#9	CLTCL1	c.1453C > G	p.Pro485Ala	Germline	NA	NA
16	#4	CUBN	c.3172A > T	p.Thr1058Ser	Germline	NA	NA
17	#10	CUBN	c.4438A > C	p.Thr1480Pro	Germline	NA	NA
18	#5	FLNB	c.3555C > A	p.Asn1185Lys	Somatic	NA	NA
19	#10	FLNB	c.3792C > A	p.Asp1264Glu	Germline	NA	Uncertain significance
20	#11	HRAS	c.182A > G	p.Gln61Arg	Somatic	likely Oncogenic	Likely pathogenic
21	#12	HRAS	c.182A > G	p.Gln61Arg	Somatic	likely Oncogenic	Likely pathogenic
22	#15	IL7R	c.332T > C	p.Val111Ala	Germline	NA	NA
23	#2	IL7R	c.460C > T	p.His154Tyr	Germline	NA	Uncertain significance
24	#12	KMT2C	c.3485_3486del	p.Lys1162SerfsTer19	Somatic	NA	NA
25	#5	KMT2C	c.11665A > C	p.Lys3889Gln	Germline	NA	NA
26	#8	KMT2C	c.12112C > T	p.Pro4038Ser	Germline	NA	NA
27	#5	LRP1B	c.11483G > T	p.Arg3828Leu	Germline	NA	NA
28	#12	LRP1B	c.10597T > C	p.Trp3533Arg	Germline	NA	NA
29	#2	NF1	c.2407C > A	p.Gln803Lys	Somatic	NA	Pathogenic
30	#8	NF1	c.3454_3455del	p.Leu1152ThrfsTer42	Somatic	NA	NA
31	#14	NF2	c.1397G > A	p.Arg466Gln	Germline	NA	Uncertain significance
32	#11	NF2	c.1439C > T	p.Thr480Met	Germline	NA	Benign
33	#14	NOTCH4	c.5522C > T	p.Pro1841Leu	Germline	NA	NA
34	#1	NOTCH4	c.1753C > T	p.Arg585Cys	Germline	NA	NA
35	#2	NOTCH4	c.3774C > A	p.Tyr1258Ter	Germline	NA	NA
36	#4	PER1	c.1421C > T	p.Pro474Leu	Somatic	NA	NA
37	#8	PER1	c.1114G > A	p.Asp372Asn	Germline	NA	NA
38	#4	PKHD1	c.3241C > T	p.Arg1081Cys	Germline	NA	Uncertain significance
39	#12	PKHD1	c.2347C > T	p.Arg783Trp	Germline	NA	Uncertain significance
40	#4	PRKDC	c.8265A > C	p.Glu2755Asp	Germline	NA	NA
41	#7	PRKDC	c.874T > C	p.Ser292Pro	Germline	NA	NA
42	#1	RET	c.2897C > T	p.Thr966Ile	Germline	NA	Uncertain significance
43	#15	RET	c.1900T > C	p.Cys634Arg	Germline	Oncogenic	Pathogenic
44	#10	SDHB	c.599G > T	p.Trp200Leu	Somatic	NA	NA
45	#13	SDHB	c.194T > C	p.Leu65Pro	Germline	NA	Uncertain significance
46	#2	SDHB	c.689G > A	p.Arg230His	Germline	NA	Pathogenic
47	#7	SDHB	c.137G > A	p.Arg46Gln	Germline	Likely Oncogenic	Likely pathogenic
48	#3	SDHB	c.725G > A	p.Arg242His	Germline	Likely Oncogenic	Pathogenic
49	#1	SETD2	c.7143dup	p.Ser2382LeufsTer47	Somatic	NA	NA
50	#4	SETD2	c.401del	p.Lys134SerfsTer18	Somatic	NA	NA
51	#1	SETD2	c.6895G > A	p.Gly2299Arg	Germline	NA	NA
52	#2	SMARCA4	c.929G > A	p.Arg310His	Somatic	NA	Uncertain significance
53	#9	SMARCA4	c.602A > T	p.Gln201Leu	Germline	NA	Likely benign
54	#12	SYNE1	c.7968C > A	p.Ser2656Arg	Somatic	NA	NA
55	#8	SYNE1	c.8686C > T	p.Arg2896Cys	Germline	NA	NA
56	#1	TP53	c.326dup	p.Arg110ProfsTer39	Somatic	NA	NA
57	#11	TP53	c.31G > C	p.Glu11Gln	Germline	NA	Uncertain significance
58	#10	TP53	c.725G > A	p.Cys242Tyr	Germline	Likely Oncogenic	Pathogenic

# Pt, patient number; NA, not available. The detail information on potential clinically actionable mutations or clinical significance was annotated using OncoKB and ClinVar database, respectively. In the case of mutations with conflicting interpretations, the latest report was considered.

## Data Availability

The data presented in this study are available in Table 2 in article and Appendix A.

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
