# Peer review of "Mutation Profile of Aggressive Pheochromocytoma and Paraganglioma with Comparison of TCGA Data"

_cancers, 2021, doi:10.3390/cancers13102389_

Round 1

Reviewer 1 Report

In the present study, Choi and co-workers analyzed the mutation profile of aggressive PPGLs using targeted NGS analysis and compared their results with that of clinically aggressive PPGLs from TCGA data to characterize the genetic events associated with the aggressive behavior including metastasis in PPGLs. In my view, the manuscript is of interest to the clinical community and summarizes well valuable information; therefore I believe it would be of interest to the readership of Cancers. However, there are some points that need to be addressed. The manuscript would benefit from major revision and further clarification with respect to it's limitations.

Reviewer's comments:

  1. As compared to TCGA data and the study published by Fishbein et al. in Cancer cell in 2017, please specify what is the novel hypothesis of the present study and what are the novel findings of it, as for example the prognostic impact of SDHB mutation is already established.
  2. In order to address the metastatic potential of PPGL, a control group of non-metastatic tumors subjected to adequate follow-up would be needed to make clinical inferences. 
  3. Along the same line, what is purpose and the actual comparisons between the AMC and the subset of aggressive tumors from the TCGA cohort, apart from TMB (Validation?) and how is this translated to a clinical meaningful message?
  4. With regards to survival analyses (OS and DFS), the sample size of the AMC cohort precludes any safe conclusions to be derived on the prognostic impact of the mutational profile of these tumors. 
  5. Along the same line, please specify length of follow-up and provide patients at risk bellow Kaplan Meyer curves in all relevant figures.
  6. In order to elucidate potential therapeutic targets and treatment response in respect of the tumor´s mutational profile in the TMA cohort please clarify treaments that these patients received and how they responded.
  7. Please elaborate more in the Conclusions Section of the study, the particular implications of the findings of this study in clinical practice with regards to targeted treatments and intensity of surveillance, as well as future research directions.

Author Response

We highly appreciate for the careful review and insightful comments about our manuscript. We believe that your comments have definitely improved our manuscript. We addressed each concern in a point-by-point fashion as follow. Changes or insertions are shown with the red color in revised manuscript.

1. As compared to TCGA data and the study published by Fishbein et al. in Cancer cell in 2017, please specify what is the novel hypothesis of the present study and what are the novel findings of it, as for example the prognostic impact of SDHB mutation is already established.

: Initially our purpose was to discover new variants and to find if any new variants may impact on prognosis. Even though we could not find prognostic impact of any new variants or new mutational profile, we think we at least added some evidence of the role of several alterations of chromatin remodeling genes including SETD2, ATRX, KMT2C and SMARCA4 through the unknown mutation profile group for the progression of PPGLs.

2. In order to address the metastatic potential of PPGL, a control group of non-metastatic tumors subjected to adequate follow-up would be needed to make clinical inferences. 

: Thank you for your comment. Although your comment is exactly true and important point, in this study we could not actually include real non-metastatic samples because we could not assure that apparently non-metastatic PPGL at present time would never reveal aggressive behavior during further follow-up period. So we chose similar aggressive patients in TCGA group instead and compared with our results. We added this point as one of study limitation in the discussion. (line 349)

3. Along the same line, what is purpose and the actual comparisons between the AMC and the subset of aggressive tumors from the TCGA cohort, apart from TMB (Validation?) and how is this translated to a clinical meaningful message?

: Thank you for your comment. As you mentioned, our cohort is too small and could not include real non-metastatic control group. Instead we tried to validate our results in the similar number of aggressive patients in TCGA cohort and revealed slightly higher total mutation burden and similar overall and disease free survival to ours.

4. With regards to survival analyses (OS and DFS), the sample size of the AMC cohort precludes any safe conclusions to be derived on the prognostic impact of the mutational profile of these tumors.

: Thank you for your comment. We additionally presented this limitation in the discussion. (line 354)

5. Along the same line, please specify length of follow-up and provide patients at risk bellow Kaplan Meyer curves in all relevant figures.

: We revised figure 5 (line284) to specify the number of patients at risk. We also made new supplementary table 2 which included the length of follow-up period for each patient.

6. In order to elucidate potential therapeutic targets and treatment response in respect of the tumor´s mutational profile in the TMA cohort please clarify treatments that these patients received and how they responded.

: We made new supplementary table 2 to present clinical information for each subject including their treatment modalities and response to individual treatment. (line 180)

7. Please elaborate more in the Conclusions Section of the study, the particular implications of the findings of this study in clinical practice with regards to targeted treatments and intensity of surveillance, as well as future research directions.

: Thank you for your comment. We elaborated more in the conclusion section in line 369.

Reviewer 2 Report

Choi et al. performed NGS in 15 aggressive PPGL to characterize the genomic abnormalities and compared it with the TCGA database. The authors concluded that those findings could be useful to determine therapeutic intervention and outcomes.

Could the authors clarify more about the germline testing? In the materials and methods is well described about the somatic testing from FFPE, however, it is not clear if germline testing was done from saliva or blood. Based on the pathogenic variants from somatic findings, did the authors confirmed with a germline panel or a NGS?

Could the authors comment about the size of the primary tumors. Size and location (adrenal or extra adrenal) are independent predictors of malignancy (Zelinka et el 2011, eisnehifer et al 2012).

We don't make medical decisions on variants of unknown significance, likely benign and if a NSG was done all of the pathogenic and likely pathogenic variants is reported. Could the authors clarify that? I personally would not make conclusions on VUS or benign and likely benign. That needs to be clarify and exclude those mutations.

The paper has it merit and it is well-written, however, a late mutation has been reported associated with more aggressive behavior (Crona and Skogseid EJE 2016). The authors needs to clarify how their paper could help improve outcomes in patients with aggressive/metastatic PPGL different to what is published. 

Author Response

We highly appreciate for the careful review and insightful comments about our manuscript. We believe that your comments have definitely improved our manuscript. We addressed each concern in a point-by-point fashion as follow. Changes or insertions are shown with the red color in revised manuscript.

1. Could the authors clarify more about the germline testing? In the materials and methods is well described about the somatic testing from FFPE, however, it is not clear if germline testing was done from saliva or blood. Based on the pathogenic variants from somatic findings, did the authors confirmed with a germline panel or a NGS?

: We performed targeted NGS by extracting genomic DNA from FFPE samples of matching normal tissues in the same way as tumor tissues. To detect germline variants, we used haplotypecaller program of gatk and manually filtered germline variants at over 50% of the mutation allele frequency. To clarify, we wrote as ‘tumor or matched normal FFPE’ in line 98.

2. Could the authors comment about the size of the primary tumors. Size and location (adrenal or extra adrenal) are independent predictors of malignancy (Zelinka et el 2011, eisnehifer et al 2012).

: We made new supplementary table 2 to present clinical information for each subject including tumor size and location. (line 180)

3. We don't make medical decisions on variants of unknown significance, likely benign and if a NSG was done all of the pathogenic and likely pathogenic variants is reported. Could the authors clarify that? I personally would not make conclusions on VUS or benign and likely benign. That needs to be clarify and exclude those mutations.

: Thank you for your comment. As your comment, it's really difficult to make medical decision on clinical significance for variation with unknown significance and likely benign.

 In general, when analyzing mutational profile using only somatic mutations, variants known to be pathogenic or likely pathogenic are used. However, since the main goal of this study is to discover new variations associated with aggressive PPGL and to determine whether these alterations affect clinical prognosis, all germline and somatic alterations were used in the analysis.

Moreover, the clinical significance of a particular mutation is often interpreted and reported differently according to various databases. Therefore, certain variants reported as VUS or likely benign in the OncoKB and ClinVar databases used in this study could be interpreted as clinically important mutations in different databases. For this reason, in this study, we think it is more appropriate to analyze all variations regardless of their clinical significance.

4. The paper has it merit and it is well-written, however, a late mutation has been reported associated with more aggressive behavior (Crona and Skogseid EJE 2016). The authors needs to clarify how their paper could help improve outcomes in patients with aggressive/metastatic PPGL different to what is published. 

: Thank you for your important comment. Our results might share some results with study by Crona et al. regarding late somatic mutation including ATRX, SETD2, KMT2C and SMARCA4 mutation in evolution of aggressive PPGLs. Our unknown mutation profile group revealed 60% genetic alterations of chromatin remodeling genes including SETD2, ATRX, KMT2C and SMARCA4. We think our results might have strength in trying to present their prognostic impact even in all aggressive PPGLs with heterogeneous prognosis. We elaborated more in the revised conclusion section in line 369.   

Reviewer 3 Report

In the article “Mutation profile of aggressive pheochromocytoma and paraganglioma with comparison of TCGA data” the authors reported the NGS data regarding fifteen aggressive PPGL compared with TCGA.

In this study 11 pheochromocytoma and 4 paragangliomas were included. The authors included only sympathetic PGL or sympathetic and parasympathetic ones?

In my opinion this aspect is relevant considering the better prognosis in patients affected by HNPGL.

For the same reason the absence of biochemical profiles is not satisfactory. The authors declared this limitation. In fact, the presence of symptoms related to elevated level of catecholamines could influence the overall survival.

The authors didn’t discuss the results of MAPP PRONO study that is the European study that evaluated prognostic parameters of OS in metastatic PHEO and PGL. 

The authors reported an increase risk of death in SDHB patient and no difference in term of DFS. 

They obtained the same result considering all patients included in the pseudohypoxia cluster. 

In MAPP PRONO study Hescot et al didn’t confirm a prognostic role of SDHB mutation. In their series some confounding factors were present such as HNPGL, absence of catecholamine excess, young age and low tumor burden. For this reason, in my opinion is important add some data regarding localization and secretion.

The authors reported that patients who had no deletion in chromosome arm 1p presented a complete response (CR). In table 1 response to therapies are also reported considering RECIST criteria. In my opinion this aspect is not clear without mention tumor burden and the different kind of therapiy used. In fact, patients who had no deletion in chromosome arm 1p could present a complete response due to genetic aspect or due to lower tumor burden or different kind of therapy. I suggest mentioning these aspects if the authors would like to maintain the RECIST results. 

In table 1 put “a” in apex at AJCC staginga

Author Response

We highly appreciate for the careful review and insightful comments about our manuscript. We believe that your comments have definitely improved our manuscript. We addressed each concern in a point-by-point fashion as follow. Changes or insertions are shown with the red color in revised manuscript.

1. In this study 11 pheochromocytoma and 4 paragangliomas were included. The authors included only sympathetic PGL or sympathetic and parasympathetic ones? In my opinion this aspect is relevant considering the better prognosis in patients affected by HNPGL.

: Thank you for your comment. In AMC cohort, we only included sympathetic PPGLs. We denoted ‘sympathetic PGL’ in line 83.

2. For the same reason the absence of biochemical profiles is not satisfactory. The authors declared this limitation. In fact, the presence of symptoms related to elevated level of catecholamines could influence the overall survival.

: For more details of clinical information including biochemical profiles, we made new supplementary table 2. (line 180) But as we have declared as one of limitation, we only could find just functioning or not for biochemical profiles. Most subjects had been treated at very earlier time with limited clinical data and did not check further fractionated hormone levels during follow-up.

3. The authors didn’t discuss the results of MAPP PRONO study that is the European study that evaluated prognostic parameters of OS in metastatic PHEO and PGL. The authors reported an increase risk of death in SDHB patient and no difference in term of DFS. They obtained the same result considering all patients included in the pseudohypoxia cluster. In MAPP PRONO study Hescot et al didn’t confirm a prognostic role of SDHB mutation. In their series some confounding factors were present such as HNPGL, absence of catecholamine excess, young age and low tumor burden. For this reason, in my opinion is important add some data regarding localization and secretion.

: Thank you for your kind comment. We reviewed the MAPP PRONO study and presented additional clinical information of AMC cohort in new supplementary table 2. We additionally discussed about this meaningful study in the discussion section. (line 312 and 325)

4. The authors reported that patients who had no deletion in chromosome arm 1p presented a complete response (CR). In table 1 response to therapies are also reported considering RECIST criteria. In my opinion this aspect is not clear without mention tumor burden and the different kind of therapy used. In fact, patients who had no deletion in chromosome arm 1p could present a complete response due to genetic aspect or due to lower tumor burden or different kind of therapy. I suggest mentioning these aspects if the authors would like to maintain the RECIST results.

: As your comment, we could not adjust numerous clinical variables including different kind of therapies and tumor mutation burden to make proper conclusion for RECIST response criteria. But in order to provide as much clinical information as possible, we additionally presented treatment modalities and various clinical data including hypersecretion or tumor size in new supplementary table 2.

5. In table 1 put “a” in apex at AJCC staging

: Thank you for your kind comment. We revised this error in table 1.

Round 2

Reviewer 1 Report

The Authors have answered adequately to my points and overall, have improved the quality of the paper.

Reviewer 2 Report

The authors answered all of my questions appropriately. I do not have further comments at this time. 

Reviewer 3 Report

I would like to thanks the authors for considering my suggestions. The addition of supplementary table 2 improves the manuscript.